# Tribological Properties of Chromia and Chromia Composite Coatings Deposited by Plasma Spraying

**Lukas Bastakys** [1], **Liutauras Marcinauskas** [1,2,*], **Mindaugas Milieška** [2], **Matas Grigaliūnas** [1], **Sebastjan Matkovič** [3] **and Mindaugas Aikas** [2]

[1] Department of Physics, Kaunas University of Technology, Studentų 50, 51368 Kaunas, Lithuania; lukas.bastakys@ktu.edu (L.B.); matulskis@gmail.com (M.G.)

[2] Plasma Processing Laboratory, Lithuanian Energy Institute, Breslaujos 3, 44403 Kaunas, Lithuania; mindaugas.milieska@lei.lt (M.M.); mindaugas.aikas@lei.lt (M.A.)

[3] Laboratory for Tribology and Interface Nanotechnology, Faculty of Mechanical Engineering, University of Ljubljana, Bogišićeva 8, 1000 Ljubljana, Slovenia; sebastjan.matkovic@tint.fs.uni-lj.si

\* Correspondence: liutauras.marcinauskas@ktu.lt; Tel.: +370-6151-0490

**Abstract:** $Cr_2O_3$ and $Cr_2O_3$–$SiO_2$-$TiO_2$ coatings are deposited on P265GH steel using atmospheric plasma spraying. The influence of silicon oxide—titanium oxide addition on the surface morphology of the coatings, phase composition and tribological properties under non-lubricated sliding conditions are investigated. The addition of $SiO_2$-$TiO_2$ led to the formation of a more uniform surface morphology and reduce the surface roughness of the $Cr_2O_3$ coatings. The X-ray diffraction (XRD) studies indicated that both coatings are composed of an eskoloite $Cr_2O_3$ phase. The friction coefficients of the $Cr_2O_3$ coating are 0.504 and 0.431 when 1 N and 3 N loads were used, respectively. Meanwhile, the $Cr_2O_3$–$SiO_2$-$TiO_2$ coating demonstrated slightly lower values of friction coefficients under similar loads. The specific wear rate of the as-sprayed coating is reduced with the addition of $SiO_2$-$TiO_2$. It was found that the wear rates of the $Cr_2O_3$ and $Cr_2O_3$–$SiO_2$-$TiO_2$ coatings are up to 20 times lower compared to the steel substrate. This article is an expanded version of the "19th international conference on plasma physics and applications" conference abstract.

**Keywords:** plasma spraying; chromia coatings; $Cr_2O_3$–$SiO_2$-$TiO_2$; tribological properties; friction coefficient



## 1. Introduction

Nowadays, chromium oxide coatings are widely used as protective coatings for metal alloys or steels due to their thermal conductivity, high corrosion resistance and excellent tribological properties [1–3]. Considering the high melting temperature (~2436 °C) of $Cr_2O_3$ material [4], various thermal spraying (high-velocity oxygen fuel, detonation spray, plasma spraying) technologies for the production of $Cr_2O_3$ coatings are used. Meanwhile, plasma spraying was chosen as the deposition method due to its relatively high deposition efficiency, high flexibility and high plasma temperatures and velocities [5].

Currently, many attempts have been made to improve the wear resistance and reduce the friction coefficient of $Cr_2O_3$ coatings. It was demonstrated that doping with $TiO_2$ [6] or $Al_2O_3$ [7] improves the tribological properties of $Cr_2O_3$ coatings. Moreover, self-lubrication by incorporating a solid lubricant such as $CaF_2$ [8], graphene nanoplatelets [9] or nano-Ag [10] effectively reduces the friction coefficient and enhances the wear resistance of $Cr_2O_3$ coatings.

Vernhes et al. [11] indicated that the friction coefficient of $Cr_2O_3$ coatings was ~0.5. Meanwhile, the coefficient of friction (COF) of $TiO_2$-$Cr_2O_3$ coatings were ranging from 0.6 to 0.7. F.L. Toma et al. [12] obtained that the wear rate and COF of $Cr_2O_3$ coatings were 0.56 (plasma sprayed) and 0.75 (HVOF) and $1.7 \times 10^{-5}$ and $3.1 \times 10^{-6}$ mm³/(Nm), respectively. The reduction of tribological properties of chromium oxide coatings with an addition of $TiO_2$ was observed in several researches [13–15]. Kiilakoski et al. [16] demonstrated that

the microstructure, surface roughness, hardness and cavitation resistance of $Cr_2O_3$ coatings strongly depended on the spraying parameters (distance, suspension flow rate, etc). Li et al. [6] observed that the addition of $TiO_2$ reduced the hardness of $Cr_2O_3$-$TiO_2$ coatings and the COF values were distributed in the range of 0.56–0.91 under dry lubrication. The authors found that the lowest COF values were obtained when the amount of $TiO_2$ was at 16%. Venturi et al. [9] obtained that the friction coefficient was reduced from 0.60 to 0.51 and the specific wear rate was decreased by 20% with the addition of graphene nanoplatelets into chromia coatings. Ratia et al. [17] observed that the wear rate of chromia coatings increased with the increase in ambient temperature. Babu et al. [18] demonstrated that the detonation-sprayed $Cr_2O_3$-20 wt% $Al_2O_3$ coating showed between three and six times improved tribological performance under dry sliding and abrasive wear modes compared to plasma-sprayed $Cr_2O_3$ coating. Yang et al. [4] obtained that the addition of $CeO_2$ or $Nb_2O_5$ into the $Cr_2O_3$ coating enhanced the average friction coefficients (for sliding durations of 1000–10,000 s) from 0.3135 ($Cr_2O_3$) to 0.3149 ($Cr_2O_3$-$CeO_2$) and 0.4414 ($Cr_2O_3$-$Nb_2O_5$), respectively.

The information about the deposition of chromia and chromia-silicon dioxide-titania coatings by air-hydrogen plasma and the tribological properties of such coatings is limited in the scientific literature. The main aim of this study was to determine the influence of $SiO_2$-$TiO_2$ addition on the surface morphology, phase composition friction coefficient and wear rate of $Cr_2O_3$ coatings formed using air-hydrogen plasma by atmospheric plasma spraying.

## 2. Materials and Methods

Chromia ($Cr_2O_3$) and chromia composite ($Cr_2O_3$-$SiO_2$-$TiO_2$) coatings were produced on P265GH steel using atmospheric plasma spraying. For chromia coatings deposition the chromium oxide (99.7%) (MOGUL PC 18, Mesh size $-45 + 22$ μm, MOGUL MET-ALLIZING GmbH, Kottingrunn, Austria) powder was used. Meanwhile, the chromia composite coatings were formed using $Cr_2O_3$-$SiO_2$-$TiO_2$ (92/5/3, MOGUL PC 17, Mesh size $-45 + 22$ μm, MOGUL METALLIZING GmbH, Kottingrunn, Austria) powder, where $SiO_2$ and $TiO_2$ were mixed in 5% and 3% by weight, respectively. Air plasma jet's total gas flow rate was 3.7 g/s. Powders were fed into the plasma jet using powder-carrier gas (air) with a flow rate of 0.48 g/s. In order to increase the melting degree of the powder particles, hydrogen (0.053 g/s) gas was used [19]. The P265GH steel was placed on the water-cooled holder. The dimensions were 40 mm $\times$ 10 mm $\times$ 6 mm. An aluminum bonding layer was used to achieve higher adhesion between the coating and the substrate. The arc current was kept constant at 180 A, which resulted in ~37.8 kW torch power. During the coating formation, the distance between the exit of the nozzle and the surface of the coating was 70 mm and spraying duration was 40 s. The mean plasma jet temperature inside the nozzle chamber (at powder injection place) was $3685 \pm 50$ K. Meanwhile, the mean plasma temperatures at the exit of the torch nozzle were $3405 \pm 50$ K and $3420 \pm 50$ K when $Cr_2O_3$ and $Cr_2O_3$-$SiO_2$-$TiO_2$ coatings were formed, respectively. The slight temperature variation during formation of coatings is due to arc discharge fluctuations and the heat exchange processes taking place in the plasma.

The surface morphology was investigated using scanning electron microscopy (SEM) Hitachi S-3400N (Hitachi, Tokyo, Japan). The elemental composition of the deposited coatings was determined by energy-dispersive X-ray spectroscopy (EDS) using Bruker Quad 5040 spectrometer (AXS Microanalysis GmbH, Billerica, MA, USA). EDS measurement data was collected from the surface area at $\times100$ magnification and at least five different surface areas were measured. The linear surface roughness was analyzed using a surface roughness tester Mitutoyo Surftest SJ-210 Series (Version 2.00 with standard ISO 1997 Mitutoyo, Kawasaki, Japan). The length of one profile measurement was 4 mm. The three samples of each type of coating were measured and at least five measurements of each sample were performed in order to calculate average linear roughness values. The phase composition of the coatings was investigated by X-ray diffraction (XRD) using Bruker D8 Discover diffractometer (Bruker D8 Discover, Billerica, MA, USA). Measurements were carried out

using standard Bragg-Brentano geometry and $CuK_\alpha$ ($\lambda$ = 0.154059 nm) radiation source. DIFFRAC.EVA software was used to process the diffraction patterns. The friction coefficient was measured using a CETR-UMT-2 ball-on-flat tribometer (CETR, Campbell, CA, USA) with 1 N and 3 N loads at dry-sliding conditions (sliding duration 120 min, sliding speed of 0.1 m/s, distance of 720 m). As the counterpart a 10 mm diameter $Al_2O_3$ ball (grade 10 and purity 99.5%) was used. The length of the stroke was 5 mm. The tribological tests on the $Cr_2O_3$ and $Cr_2O_3$–$SiO_2$-$TiO_2$ coatings were performed three times for two samples of each series. Specific wear rates were estimated from the surface profiles, which were measured using Ambios XP-200 Profiler (Ambios Technology Inc., Santa Cruz, CA, USA).

## 3. Results and Discussion

SEM images of as-sprayed $Cr_2O_3$ and $Cr_2O_3$-$SiO_2$-$TiO_2$ coatings are presented in Figure 1. The $Cr_2O_3$ coating surface (Figure 1a,b) consists of fully and partly melted particles, which results in a relatively dense lamellar structure. Higher magnification images (Figure 1b) reveal the formation of pores, splats and micro-cracks, which could result from the temporary thermal expansion of steel during the deposition. During the deposition, the substrate temperature increases, which causes it to expand and increase the deposition area. Afterward, rapid cooling causes steel to shrink and induces micro-stresses in the coating.

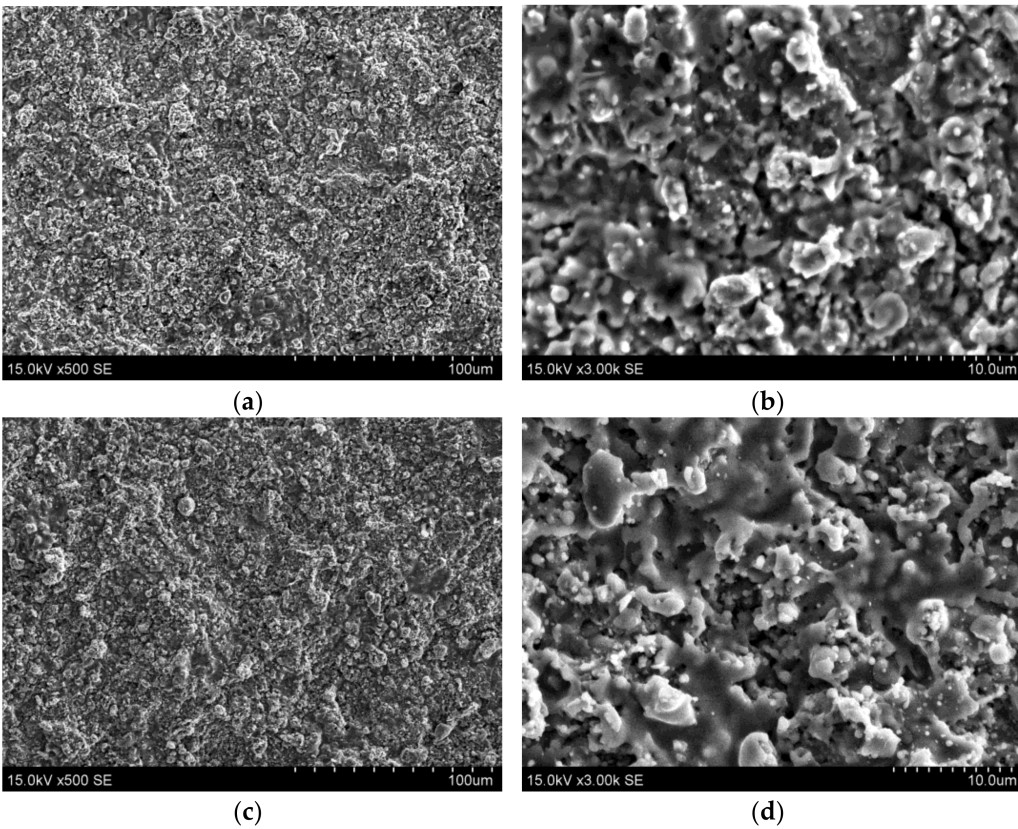

**Figure 1.** Surface morphology of (**a,b**) $Cr_2O_3$ and (**c,d**) $Cr_2O_3$-$SiO_2$-$TiO_2$ coatings.

A SEM surface image of the $Cr_2O_3$-$SiO_2$-$TiO_2$ coating (Figure 1c) shows no obvious difference in the surface morphology. However, at a higher magnification view (Figure 1d), it is seen that the number of splats increased on the surface of the sprayed $Cr_2O_3$-$SiO_2$-$TiO_2$ coating, compared to the $Cr_2O_3$ coating. The boundary lines of flat zones have a clear image, and there are some micro-holes distributed along those lines. The porosity of plasma sprayed coatings is associated with the existence of coarse and fine pores. Coarse porosity is related to structural defects due to incomplete filling of voids between previously impacting feedstock particles. Meanwhile, fine pores are formed due to the release of gases from

the molten splats during cooling [15]. Zamani et al. [7] indicated that the porosity of the $Cr_2O_3$ coatings was ~4.0%, and the porosity was reduced with the increase in the $Al_2O_3$ amount in the sprayed coatings. Singh et al. [15] demonstrated that the average porosity of the $Cr_2O_3$–3%$TiO_2$ coating sprayed using a conventional powder was ~3.9%. Yang et al. [4] observed that the porosity of the $Cr_2O_3$ coating was 6.9%. The addition of 2 wt.% of $CeO_2$ or $Nb_2O_5$ reduced the porosities of coatings to 4.8% and 5.0%, respectively [4]. Mao et al. [20] indicated that the porosity of $Cr_2O_3$ coating was 15.8%. The porosity of coatings was reduced from 7.5% to 4.3% with the addition of various amounts of $Al_2O_3$ into $Cr_2O_3$. It is expected that the addition of $SiO_2$ and $TiO_2$ will slightly reduce the porosity of the $Cr_2O_3$ coating. The surface morphology images indicate that the chosen plasma spraying parameters are applicable for chromia and chromia composite coating formation.

The EDS results demonstrated that the $Cr_2O_3$ coating consisted of chromium (~76.4 wt%) and oxygen (~22.8 wt%) with a low amount (less than 0.3 wt%) of impurities related to the composition of the powder. The $Cr_2O_3$-$SiO_2$-$TiO_2$ coating is composed of chromium (~73.9 wt%), oxygen (~21.5 wt%), titanium ~3.4 wt%) and silicon (~0.7 wt%). It should be noted that a low amount of carbon (up to 0.5 wt.%) was obtained on the surface of both coatings due to the absorption of atmospheric gases. The EDS measurements indicated that the distribution of the chromium, oxygen, titanium and silicon on the sprayed $Cr_2O_3$-$SiO_2$-$TiO_2$ coating surface is uniform and homogenous. The surface roughness ($R_a$) of the $Cr_2O_3$ coating was $3.31 \pm 0.29$ μm, while the root-mean-square roughness ($R_q$) was $4.15 \pm 0.43$ μm. The addition of the $TiO_2$ and $SiO_2$ reduced the surface roughness values ($R_a$ was $2.22 \pm 0.36$ μm and $R_q$ was $2.76 \pm 0.45$ μm). The decrease in surface roughness of the $Cr_2O_3$-$TiO_2$-$SiO_2$ coating in comparison to the chromia could be related to the lower melting temperatures of $TiO_2$ and $SiO_2$ materials. The melting temperature of $Cr_2O_3$ is ~2710 K, while the melting temperatures of $TiO_2$ and $SiO_2$ are ~2115 K and ~1985 K, respectively [4]. The mean plasma jet temperature in the feedstock powder injection place was about 3685 K, while at the exit of the torch nozzle it was 3420 K (for $Cr_2O_3$-$SiO_2$-$TiO_2$) and 3405 K (for $Cr_2O_3$), respectively. Thus, the melting degree of the $Cr_2O_3$-$SiO_2$-$TiO_2$ particles is slightly higher compared to the $Cr_2O_3$ particles. The feedstock powder particles, as a result of the impacting effect of plasma flow while flowing through the plasma plume, reach the steel substrate in a molten or semi-molten state. The appearance of splats on the surface is a result of the splashing of molten particles after incidence on the substrate [20]. It was demonstrated that the splats start to dominate on the surface of sprayed coatings with the increase in the torch power or addition of the materials with lower melting temperature into feedstock powders [4,20–22].

The XRD analysis of the coatings indicated that no new phases appeared during the spraying compared to the initial feedstock powder (Figure 2). All peaks are identical for both patterns; the difference in the intensity of some peaks is due to the orientation of the coating and the nature of the used $Cr_2O_3$ and $Cr_2O_3$-$SiO_2$-$TiO_2$ powders. It can be seen that the $Cr_2O_3$ and $Cr_2O_3$-$SiO_2$-$TiO_2$ coatings contained only an eskolaite phase. Li et al. [6] obtained that the $Cr_2O_3$-$TiO_2$ composite coatings were composed of the eskolaite, rutile ($TiO_2$) and $(Cr_{0.88}Ti_{0.12})_2O_3$ phases. The authors indicated that the $TiO_2$ peaks were located at 27.5, 41.3, 54.4 and 56.7 at 2 theta degrees and the intensities of the peaks increased with the increase of the $TiO_2$ content from 8 to 32 wt.%. The low-intensity peaks of titania were observed in $Cr_2O_3$-3wt.%$TiO_2$ coatings prepared using conventional and nanostructured powders [15]. Figure 2 shows the XRD patterns for the formed chromia and chromia composite coatings. The detected peaks are at 24.6, 33.7, 36.3, 41.6, 50.3, 55.0, 63.5, 65.2 and 73.1 2θ degree angles, which correspond to (012), (104), (110), (202), (024), (116), (214), (300) and (1010) orientations, respectively [6,15,23]. However, the $TiO_2$ phase was not observed in the XRD patterns of the as-sprayed $Cr_2O_3$-$SiO_2$-$TiO_2$ coating. Wang et al. [23] demonstrated that the XRD analysis cannot detect components with low (less than 5%) concentration and also did not obtain the titania peaks in the XRD patterns of nano-$TiO_2$/micro-size $Cr_2O_3$ composite particles.

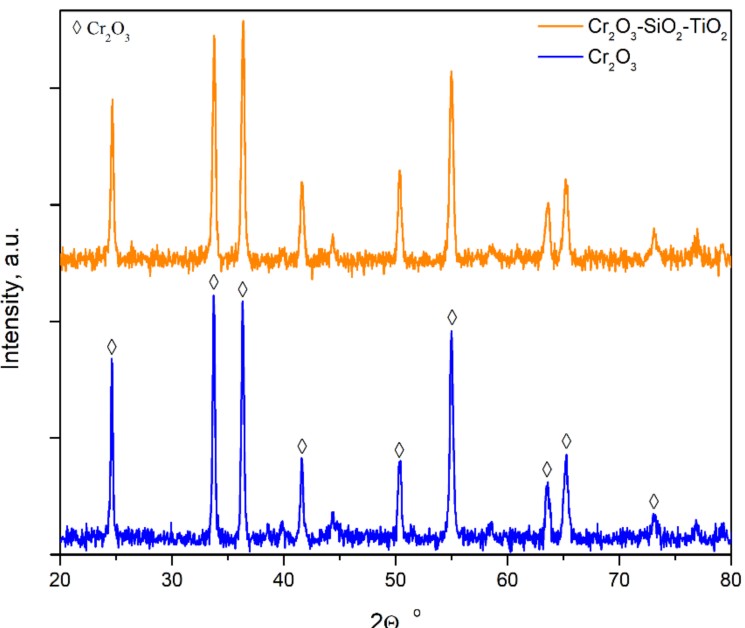

**Figure 2.** The XRD patterns of the $Cr_2O_3$ and $Cr_2O_3$-$SiO_2$-$TiO_2$ coatings.

The variation of the friction coefficient of coatings and steel versus sliding time is presented in Figure 3. The friction coefficient of the steel surface increased and stabilized at ∼0.720 when a load of 1 N was used (Figure 3a). The further increase in sliding time resulted in a marginal change in the steel COF value. The steady-state (average value of the last 10%) COF of the steel was ~0.687. For the $Cr_2O_3$ and $Cr_2O_3$-$SiO_2$-$TiO_2$ coatings, the friction coefficient is enhanced to a maximum value followed by a plateau region [24]. The existence of hills in the friction curve of $Cr_2O_3$ coating is associated with the upcoming of $Cr_2O_3$ particles in the contact area between coating and $Al_2O_3$ ball (Figure 3a). The peeled-off $Cr_2O_3$ particles acted as abrasives and increased the friction coefficient [25]. It was found that the steady-state COF of the $Cr_2O_3$ and $Cr_2O_3$-$SiO_2$-$TiO_2$ coatings under 1 N load was 0.504 and 0.327, respectively (Figure 4a). Meanwhile, the variation of friction coefficients of the as-sprayed coatings and steel under 3 N loads is presented in Figure 3b. It could be seen that the shapes of the friction coefficient curves are similar under both loads. However, the COF of steel was slightly reduced from 0.687 to 0.630 with the increase in load (Figure 4a). Meanwhile, the friction coefficients of $Cr_2O_3$ and $Cr_2O_3$-$SiO_2$-$TiO_2$ coatings were almost similar, 0.431 and 0.406, respectively (Figures 3b and 4a).

The SEM views of the worn surfaces of the deposited coatings under different loads are given in Figure 5. The wear track images indicated that only top hills were slightly removed from the surface when 1 N load was used (Figure 5a,c). The surface profilometry measurements indicated that it was not possible to determine the specific wear rate values of coatings as only insignificant wear was obtained. The shape of the surface profile after the friction tests was identical to the initial surface roughness profiles. The images of the coatings after the tribological tests with 3 N loads demonstrated that the damage level of the coatings was higher. The wear scars were brighter, more pronounced and had more worn-out zones compared to the wear scars obtained with 1 N load (Figure 5).

It was obtained that the specific wear rate of the steel was ~$4.22 \times 10^{-5}$ $mm^3/(Nm)$ under dry lubrication conditions with a 3 N load (Figure 4b). The specific wear rate of the chromia coating was ~$2.69 \times 10^{-6}$ $mm^3/(Nm)$. Meanwhile, the addition of a low amount of $SiO_2$ and $TiO_2$ into the $Cr_2O_2$ feedstock powers leads to the reduction of the wear rate of the coating. It was obtained that the wear rate of chromia composite coating was $1.54 \times 10^{-6}$ $mm^3/(Nm)$, which is more than 25 times lower compared to steel and almost twice lower compared to chromia coating.

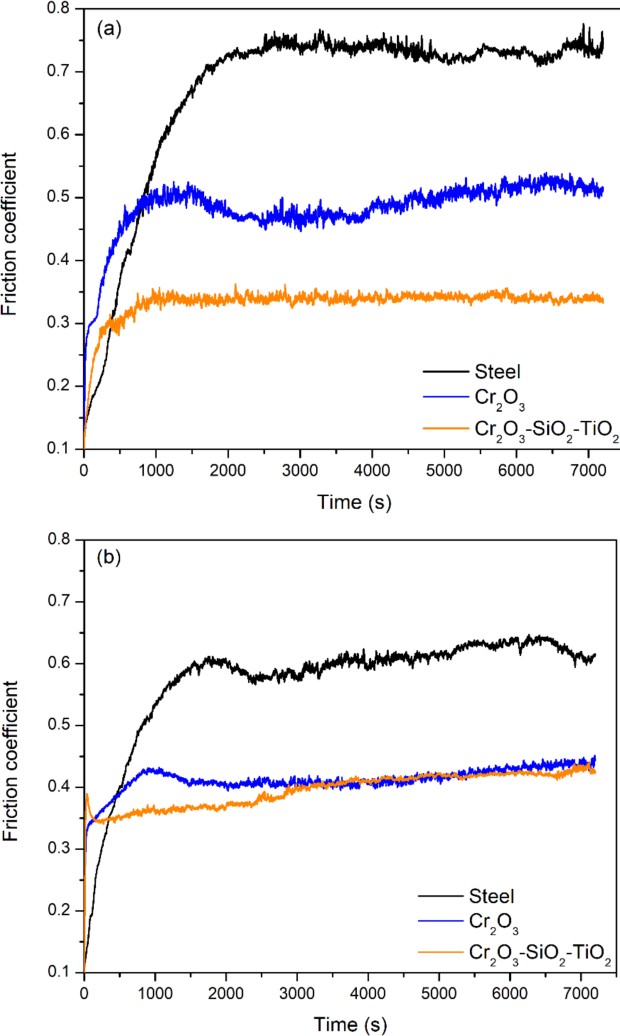

**Figure 3.** The variation of friction coefficients versus sliding time when (**a**) 1 N and (**b**) 3 N loads were used.

Ceramic coatings are considered brittle materials; thus, plastic deformation is one of the dominant wear mechanisms [25,26]. It was demonstrated that the decrease in the fracture toughness, the increase in the porosity or the number of unmolten particles in the coatings usually deteriorates the wear resistance properties of the coatings [26,27]. The wear debris of the $Cr_2O_3$ coating was observed in the shape of tiny and angular particles, probably due to its brittle fracture. Meanwhile, the smaller sized particles and plastically deformed regions were more clearly seen in the surface images of the $Cr_2O_3$–$SiO_2$-$TiO_2$ coating. The increase in sliding wear rates of $Cr_2O_3$ coatings with the addition of $TiO_2$ compared to $Cr_2O_3$ coatings during the tribological tests against an $Al_2O_3$ counterpart was reported by several authors [12–16]. It was demonstrated that the addition of $TiO_2$ reduced the hardness values and, as a result, the wear resistance of the $Cr_2O_3$-$TiO_2$ coatings was reduced [12]. However, we demonstrated that the friction coefficient and specific wear rate of $Cr_2O_3$–$SiO_2$-$TiO_2$ were reduced. Probably, the low number of additives improves the cohesion between individual splats and does not reduce the hardness of the coatings [12,19]. On the surfaces of the $Cr_2O_3$ and $Cr_2O_3$–$SiO_2$-$TiO_2$ coatings after tribological tests, traces of aluminum from the alumina ball were observed. The EDS results indicated a small amount of Al in the wear tracks of coatings. The concentration of Al was ~0.5 wt.% and ~0.8 wt.% for the $Cr_2O_3$ and $Cr_2O_3$–$SiO_2$-$TiO_2$ coatings, respectively.

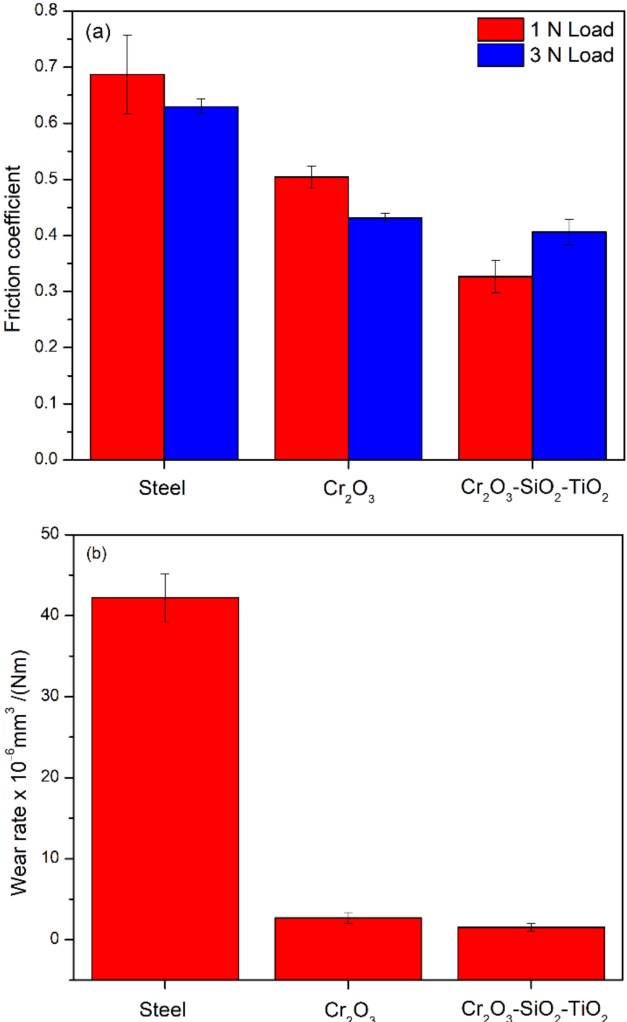

**Figure 4.** Steady-state friction coefficient values (**a**) and specific wear rates when 3 N load was applied (**b**).

Wear severity of the as-sprayed $Cr_2O_3$ coating was higher compared to the $Cr_2O_3$–$SiO_2$-$TiO_2$ coating, as shown in Figure 4b. It should be noted that the abrasive and oxidative wear areas on the worn surface of both coatings were obtained (see Figure 5a–d). Small particles associated with wear debris were also observed on the surfaces of the deposited $Cr_2O_3$ and $Cr_2O_3$–$SiO_2$-$TiO_2$ coatings. The scratches on the wear tracks of the surface of the sprayed coatings demonstrate abrasion [27]. It was demonstrated that the abrasive and oxidative wear of the as-sprayed ceramic coatings originated from the contact of high asperities, which led to the formation of small, flattened areas with a low level of oxidation [27]. Bolelli et al. [16] found that the addition of 10% $ZrO_2$, 20% $ZrO_2$ or 25% $TiO_2$ into $Cr_2O_3$ increased the specific wear rates and friction coefficient values compared to the pure $Cr_2O_3$ coating. Toma et al. [12] demonstrated higher sliding wear rates of $Cr_2O_3$–15%$TiO_2$ coatings compared to $Cr_2O_3$ coatings when an $Al_2O_3$ ball was used as a counterpart. The authors demonstrated that harder coatings showed lower friction coefficients and wear resistance values [12]. According to the authors, the ploughing regime and the crack and powder formation mode are associated with the release of micrometric-sized debris in ceramic coatings. The debris particles on the surface of our coatings after tribological tests were also obtained.

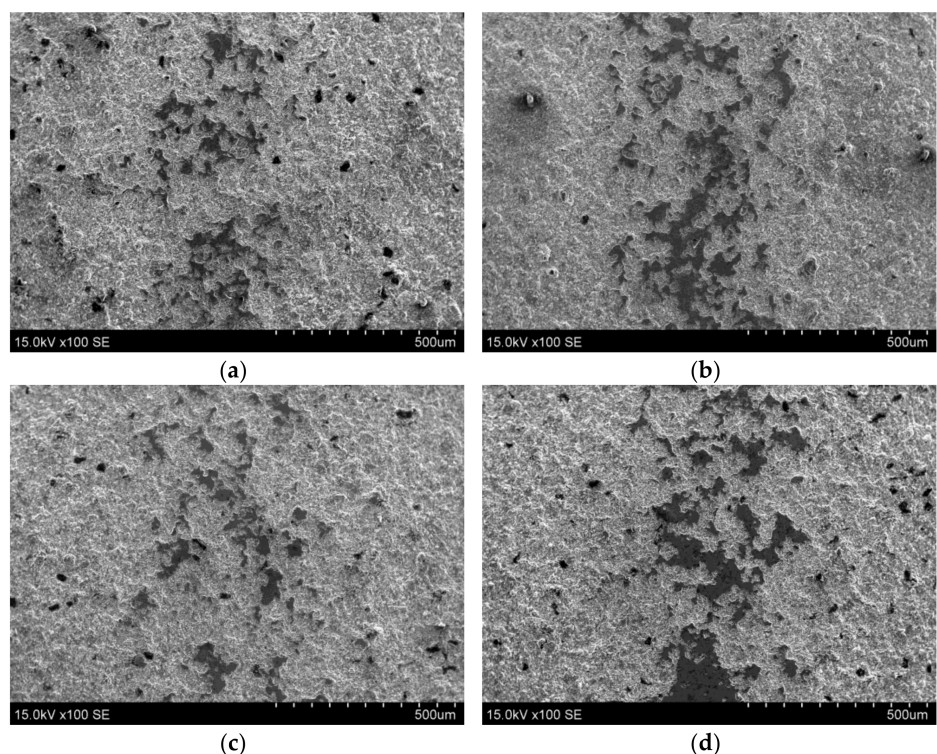

**Figure 5.** Wear scars images of (**a**,**b**) $Cr_2O_3$ and (**c**,**d**) $Cr_2O_3$-$SiO_2$-$TiO_2$ coatings after tribological test with (**a**,**c**) 1 N and (**b**,**d**) 3 N loads.

## 4. Conclusions

$Cr_2O_3$ and $Cr_2O_3$–$SiO_2$-$TiO_2$ coatings were formed by atmospheric plasma spraying using an air-hydrogen plasma. It was demonstrated that with the addition of $SiO_2$-$TiO_2$, the average surface roughness of the as-sprayed coating was reduced by up to 33%. The EDS results demonstrated that the $Cr_2O_3$-$TiO_2$-$SiO_2$ coating consisted of chromium (~73.9 wt%), oxygen (~21.5 wt%), titanium (~3.4 wt%) and silicon (~0.7 wt%). The fraction of chromium and oxygen in the $Cr_2O_3$ coating was ~76.4 wt% and ~22.8 wt%, respectively. The $Cr_2O_3$ and $Cr_2O_3$–$SiO_2$-$TiO_2$ composite coatings were composed of the eskolaite phase. It was impossible to determine the specific wear rate of the as-sprayed coatings when tribological tests were performed using a 1 N load. Only the top hills were slightly removed from the surface and the coating was insignificantly damaged after the tribological tests. Meanwhile, the specific wear rate of the steel substrate was ~$4.22 \times 10^{-5}$ $mm^3$/(Nm). The friction coefficient was slightly reduced from 0.431 to 0.406, while the specific wear rate decreased from $2.69 \times 10^{-6}$ $mm^3$/(Nm) to $1.54 \times 10^{-6}$ $mm^3$/(Nm) with the addition of $SiO_2$-$TiO_2$ into the $Cr_2O_3$ coating when 3 N load was applied.

**Author Contributions:** Conceptualization, L.B. and L.M.; methodology, L.B., M.M., M.G. and M.A..; software, L.B.; validation, L.B., M.G.; formal analysis, L.B., M.G., L.M. and S.M.; investigation, L.B., L.M., M.G., M.M., S.M. and M.A.; data curation, L.B.; writing—original draft preparation, L.B. and L.M.; writing—review and editing, L.B., S.M. and L.M.; visualization, L.B. and L.M.; supervision, L.M.; All authors have read and agreed to the published version of the manuscript.

**Funding:** This research received no external funding.

**Institutional Review Board Statement:** Not applicable.

**Informed Consent Statement:** Not applicable.

**Data Availability Statement:** Not applicable.

**Conflicts of Interest:** The authors declare no conflict of interest.

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
