# Peer review of "Tribological Properties of Chromia and Chromia Composite Coatings Deposited by Plasma Spraying"

_coatings, doi:10.3390/coatings12071035_

Round 1

Reviewer 1 Report

“Tribological properties of chromia and chromia composite coatings deposited by plasma spraying”

Authors: Lukas Bastakys, Liutauras Marcinauskas, Mindaugas Milieška, Matas Grigaliūnas, Sebastjan Matkovič and Mindaugas Aikas

The paper relates to the deposition of chromia and chromia composite coatings by plasma spraying. In introduction, the reported literature is limited to that relating to the preparation of these coatings only by this method. There is no literature (references) concerning the preparation of these coatings using other techniques. The following questions also arise:

1.    Was the surface roughness (RA, Rq) measured at only one point of the sample? Were the maximum values of RA and Rq given?

2.    What was the thickness of the obtained coatings?

3.    Have the coatings been deposited under other conditions? How change of the deposition conditions would affect the microstructure and tribological properties of obtained coatings?

4.    What about adhesion and hardness (microhardness) of the deposited coatings?

Reviewer 2 Report

The work deals with the surface coating of P265GH steel via atmospheric plasma spraying to optimise tribological properties under non-lubricated conditions. The authors used two different coating materials to compare : Cr2O3 and Cr2O3-SiO2-TiO2 composite. The effects of SiO2 and TiO2 addition to the Cr2O3 on wear rates and friction coefficient were investigated. The manuscript is well designed with experimental data supported with results. It can be published after a minor revision. 

C1) Introduction : “L. Vernhes et al. [11] indicated that the friction coefficient of Cr2O3 coatings was ~0.5. Meanwhile, The COF of TiO2- Cr2O3 coatings were ranging from 0.6 to 0.7. F.L. “ In this sentence, “COF” abbreviation is used for the first time, therefore it is supposed to explain what it stands for. 

C2) Results and discussion : “The mean plasma jet temperature in the feedstock powder injection place was about 3700 K, while at the exit of the torch nozzle was 3445 K and 3420 K, respectively.“. The nozzle temperature given as 3420 K earlier, and 3445 K value is not clearly defined here. The exit of the nozzle temperature was 3445 and 3420 K, respectively, and the two parts of the nozzle had different temperatures. Please explain it. 

C3) Results and discussion : “However, we demonstrated that the friction coefficient and specific war rate of Cr2O3–SiO2-TiO2 was reduced.” The war, supposed to be wear in this sentence, please correct it. 

Reviewer 3 Report

The authors performed the studies on "Tribological properties of chromia and chromia composite coatings deposited by plasma spraying". The article is good and informative. Authors can incorporate the following comments.

Introduction Section, Line no 2 metal allows or steels replaced with metal alloys and steels.

L. Vernhes et al. [11] need to be replaced with Vernhes et al. [11], follow  the same for other references 6, 9, 16,17 and 18

In the literature gaps, the authors mentioned that no study was reported on the chromia and chromia-silicon dioxide-titania coatings by air-hydrogen plasma and the tribological properties. Is there any industrial need to perform such a study? Why did the authors carry out this study? Who will benefit from this study?

What is the porosity of the coating in the present study?

What is the coating thickness in the present study ?. Because coating thickness plays a major role in the properties.

The authors can refer to the following articles, which give an insight into the coating.

1.       Performance of Plasma spray coatings on Inconel 625 in Air oxidation and molten salt environment at 800oC”, International Journal of ChemTech Research Vol.6, No.5, 2014, pp 2744-2749.

Round 2

Reviewer 1 Report

The paper in this form can be published.

Reviewer 3 Report

No comments